# Traceability of Functional Volatile Compounds Generated on Inoculated Cocoa Fermentation and Its Potential Health Benefits

**DOI:** 10.3390/nu11040884

**Published:** 2019-04-19

**Authors:** Jatziri Mota-Gutierrez, Letricia Barbosa-Pereira, Ilario Ferrocino, Luca Cocolin

**Affiliations:** 1Department of Agricultural, Forestry, and Food Science, University of Turin, Largo Paolo Braccini 2, 10095 Grugliasco, Torino, Italy; jatziri.motagutierrez@unito.it (J.M.-G.); letricia.barbosapereira@unito.it (L.B.-P.); ilario.ferrocino@unito.it (I.F.); 2Department of Analytical Chemistry, Nutrition and Food Science, Faculty of Pharmacy, University of Santiago de Compostela, 15782 Santigo de Compostela, Spain

**Keywords:** fermentation, functional volatile compounds, starter culture, yeast, roasting, chocolate, cocoa beans

## Abstract

Microbial communities are responsible for the unique functional properties of chocolate. During microbial growth, several antimicrobial and antioxidant metabolites are produced and can influence human wellbeing. In the last decades, the use of starter cultures in cocoa fermentation has been pushed to improve nutritional value, quality, and the overall product safety. However, it must be noted that unpredictable changes in cocoa flavor have been reported between the different strains from the same species used as a starter, causing a loss of desirable notes and flavors. Thus, the importance of an accurate selection of the starter cultures based on the biogenic effect to complement and optimize chocolate quality has become a major interest for the chocolate industry. This paper aimed to review the microbial communities identified from spontaneous cocoa fermentations and focused on the yeast starter strains used in cocoa beans and their sensorial and flavor profile. The potential compounds that could have health-promoting benefits like limonene, benzaldehyde, 2-phenylethanol, 2-methylbutanal, phenylacetaldehyde, and 2-phenylethyl acetate were also evaluated as their presence remained constant after roasting. Further research is needed to highlight the future perspectives of microbial volatile compounds as biomarkers to warrant food quality and safety.

## 1. Introduction

Certainly, people have been changing their food consumption patterns and lifestyle over the last decade [1]. To counteract unhealthy food choices, functional food has emerged as a strategy to increase the consciousness of the relationship between diet and disease/health to consumers. The objective of a successful functional food is to target a specific group of consumers and to meet their health demands without compromising flavor, taste, and color. In this context, the most used bioactive compounds in the food industry include alkaloids, anthocyanins, carotenoids, flavonoids, glucosinolates, isoflavones, phenolic acids, hydrolysate proteins, tannins, and phytochemical terpenes [2].

Volatile organic compounds (VOCs) are organic molecules that include esters, alcohols, aldehydes, ketones, phenols, terpenes, etc. These VOCs are synthesized naturally by a broad number of plants or microorganisms (as secondary metabolites) to enable interactions with their environment. In addition, it has been demonstrated that VOCs provide health benefits to consumers [3]. Health benefits provided by the microbial communities can be either direct or indirect. The difference between these two concepts is the ingestion of a live microorganism (direct) or the ingestion of microbial metabolites (indirect or biogenic effect) [4]. Undeniably, a biogenic effect is commonly observed in fermented foods such as chocolate.

The production of microbial metabolites in cocoa beans begins during fermentation. In this process, microorganisms, encompassing bacteria and yeasts, serve to confer taste, texture, and desirable aromas to the final product. An effective cocoa fermentation develops when a correct microbial succession of yeasts, lactic acid bacteria (LAB), and acetic acid bacteria (AAB) takes place [5,6]. The success of these dynamics is due to the nutrient content of the cocoa pulp that is used as an optimal substrate for the microbial growth, and yeasts are considered the first microorganisms growing at the beginning of the fermentation process, producing ethanol, organic acids and VOCs, that contribute as precursors of chocolate flavor [7]. For those reasons, yeasts have been widely used as starter cultures in cocoa beans with the aim to enrich the sensorial quality of chocolate. However, the modulation of the remarkable complexity of microbial communities in cocoa beans to obtain an optimal flavor fingerprinting as well as understanding the metabolic and regulatory networks concerning the production of secondary metabolites are still not clear. In this context, the present review aims to describe the development of the microbes in fermented cocoa beans, and to evaluate the individual capacities of yeast species to form aroma compounds to enhance flavor perception and nutritional or healthy values. More importantly, it assesses the most frequently identified VOCs during the three different steps of chocolate elaboration, including fermentation, roasting, and the final product, chocolate (Figure 1). It is important to clarify that the VOCs identified in the fermentation and final product were the most frequently identified VOCs in inoculated cocoa beans with yeast, while the most frequently identified VOCs during roasting were assessed from non-inoculated cocoa beans.

## 2. Microbial Composition of Fermented Cocoa Beans

The fermentation step is considered a key stage that influences the flavor potential of cocoa beans. Existing scientific data shows the complexity of the composition of the microbial population on fermented cocoa beans, that varies depending on: plant variety; environmental conditions; post-harvesting processing; type of fermentation; and agricultural practices [6,8,9,10]. The bacteria population often present during cocoa fermentation are mainly composed by LAB mostly belonging to the *Lactobacillus* and *Leuconostoc* genera, as well as AAB such as members of the genus *Acetobacter* [11,12,13,14,15,16,17,18,19,20,21,22,23,24]. In addition, some species belonging to *Bacillus* have also been rarely isolated from fermented cocoa beans. Despite the lower complexity of the bacteria population in the cocoa-fermented system, several yeasts have been identified including species belonging to *Candida, Debaromyces, Geotrichum, Hanseniaspora, Kluyveromyces, Pichia, Saccharomyces, Rhodotorula, Saccharomycopsis*, and *Wickerhamomyces* [6,12,14,15,17,18,20,21,22,23,24,25,26,27,28,29,30,31]. Besides yeast species, filamentous fungi belonging to *Aspergillus, Mucor, Neurospora, Penicillium*, and *Rhizopus* are also often reported [6,23]. Interestingly, there are some discrepancies between the relative abundance of microbial communities reported in fermented cocoa beans from different origins and fermented from different types of fermentations (box, heap). Nonetheless, a recent study suggested that not only the most abundant microbial species could affect the production of organic molecules, but also rare species were involved [6].

### 2.1. Yeasts Species Used as Starter during Cocoa Fermentation 

Some yeasts and fungi are considered a safe source of ingredients and additives for food processing because they have a positive image with consumers [32]. The interest in yeasts as starter cultures has arisen in recent years especially in relation to the addition of *Saccharomyces, Pichia, Kluyveromyces, Candida*, and *Torulaspora* to fermented cocoa beans [6,25,28,30,31,33,34,35,36,37]. Yeasts that are being used to ferment cocoa beans are shown in Table 1. It should be noted that starter cultures used to drive cocoa fermentation processes have been applied only in a few cocoa-producing countries such as Brazil, Malaysia, Indonesia, and Cameroon [6,25,28,30,31,33,34,35,36,38]. The importance of standardized cocoa fermentation process has become controversial, considering that the environmental conditions are difficult to control in most of the cocoa-producing countries. Therefore, the choice of selecting cocoa fermenting starters, discriminated based on origin and microbial communities, should be a logical choice from the available options.

The most frequently used yeast culture in fermented cocoa beans is *Saccharomyces cerevisiae*. This yeast has the capability to assimilate and ferment reducing sugars and citric acid, produce aroma substances and killer-like toxins, and it has a high pectinolytic activity and can prevent microbial pathogen growth [5,7,10,39,40,41,42]. Despite the well-known *Saccharomyces*, non-*Saccharomyces* yeast (*Kluyveromyces, Hanseniaspora, Pichia*, and *Torulaspora)* have also shown a relevant pectinolytic activity and increased the aroma complexity in wine [43,44]. However, these species exhibit a lower ethanol yield, and sugar consumption compared to *S. cerevisiae* [45]. Regardless of this characteristic, several studies have used mixed yeast cultures to inoculate fermented cocoa beans (Table 1) [6,31,46]. However, the combination of different yeast species often results in unpredictable compounds produced and/or different microbial communities, which can affect both the chemical and ecological population of fermented cocoa beans. The unpredictable changes, specifically from the ecological point of view, might be explained by the antagonistic ability of some yeast, such as *S. cerevisiae*, to inhibit the growth of non-*Saccharomyces* species (*Hanseniaspora guilliermondii, Torulaspora delbrueckii, Kluveromyces marxianus*, and *Lachancea thermotolerants)* by the production of antimicrobial peptides with a 4.0, 4.5, and 6.0 kDa [47]. Therefore, the selection of starter cultures to produce chocolate plays an important role not only in the modulation of the microbial communities, but rather to achieve optimal sensorial properties, such as cocoa, malty, and fruity flavors [6,31,48]. To this regard, future research is needed to elucidate the variability at the strain level that contributes an added value to the cocoa fermentation [25]. 

### 2.2. Quality Evaluation of the Chocolate Produce from Inoculated Cocoa Beans

Contrasting findings on the sensory analysis of chocolate produced from cocoa beans inoculated with yeasts has been recently assessed (Table 1) [28,30,31,34,38]. In detail, the consumer panel from Brazil and Malaysia described chocolates inoculated with *K. marxianus* (Brazil), *S. cerevisiae* (Malayzia and Brazil), *T. delbrueckii,* and a mixed culture of *S. cerevisiae* and *T. delbrueckii* (Brazil) with better desirable notes, flavor attributes, and global acceptability compared with chocolate produced from spontaneous cocoa bean fermentation [28,31,38]. In contrast, coffee and sour attributes with a worse acceptance were described from the chocolate produced in Brazil inoculated with a mixture of three yeast starters (*S. cerevisiae, P. kluyveri,* and *H. uvarum*) during cocoa fermentation [30]. Interestingly, the chocolate produced from different cocoa varieties originated from Brazil inoculated with *S. cerevisiae* during fermentation were clearly discriminated based on the perceptible attributes of each variety [31,34]. However, the lack of the small number of published studies regarding the sensory analysis of chocolate produced from inoculated cocoa beans with yeast species during fermentation from different countries to improve sensorial attributes are not conclusive. 

## 3. Changes in the Nutrient Composition from Fermented to Roasted Cocoa Beans

The transformation of the nutrient content of cocoa beans during fermentation plays an important role in the development of selected attributes in the final product (chocolate). Fats, proteins, and carbohydrates are the main macronutrients found in cocoa seeds (Table 2) [49,50,51,52,53]. Beans also contain amines that are already present in the unfermented dried cocoa and as expected their amount increased after fermentation and decreased after thermal cocoa processing [54,55]. The first step in processing cocoa beans is to ferment amino acids and oligopeptides, and reduce sugars (Table 2). This step is crucial for the development of the quality cocoa flavor that depends on the balance of organic compounds. In general, the biochemical processes involved over the fermentation and roasting of cocoa beans comprise the hydrolysis of sucrose and proteins, oxidation and hydrolysis of phenolic compounds, biosynthesis of alkaloids, amino acids, release of alcohols (that are also oxidized into acetic and lactic acid), and the breakdown of fatty acids [6,49,50,56,57]. 

### 3.1. Composition of Volatile Compounds from Cocoa Beans

More than 600 different VOCs have been identified in chocolate flavor. Substances such as aliphatic esters, polyphenols, unsaturated aromatic carbonyls, diketopiperazines, pyrazines, and theobromine are developed, and these compounds provide the characteristic chocolate flavor [49].

#### 3.1.1. VOCs Associated with Inoculated Cocoa Beans

A total of twenty VOCs profiles from inoculated cocoa beans with yeast starters (*n* = 10) and from chocolate produced from inoculated cocoa beans also with yeasts (*n* = 10) has been recently reported from ten different cocoa varieties using eleven different yeast strains over the world (Table 1). The identified VOCs from five different cocoa varieties inoculated with different yeasts during fermentation originated from Cameroon and Brazil were used to create a list of all the identified compounds. The data were treated as dummy variables indicating whether the VOCs were identified and only the most frequently reported were used to increase our knowledge of the probable VOCs formed when cocoa beans are inoculated with yeasts [6,25,33]. As expected, esters, alcohols, and aldehydes were the three major VOCs groups characterized in fermented cocoa beans inoculated with yeast (Figure 2).

In detail, the most predominant VOCs among the three studies were ethyl acetate, benzaldehyde, hexanoic acid, and the key aromatic markers for chocolate (2-heptanol, 2-phenylethanol, 2-phenylethyl acetate, and phenylacetaldehyde) [48], while the most abundant compounds at the end of the fermentation were ethyl octanoate, 1-butanol, 1-pentanol, phenylacetaldehyde, ethyl acetate, isoamyl acetate, limonene, and acetic acid (Table 3) [6,25,33]. It is important to highlight that recently, it has been demonstrated that the volatilome profile of cocoa beans fermented in boxes increased the production of alcohols and esters compared to heap fermentations [6]. However, not only the type of fermentation could influence the volatilome profile. It has been shown that the effect of the yeast starter on different cocoa varieties also influences the relative percentage of VOCs, such as 2-phenylethanol and ethyl acetate [33]. Concerning the dynamics of VOCs, it has been reported that the concentrations of limonene-epoxide and 1-butanol decreased over the fermentation time, while ethyl acetate, limonene, benzaldehyde, benzyl alcohol, acetoin, 3-methyl-1-butanol, acetic acid, and the key-aroma markers (phenylacetaldehyde, 2-heptanol and, 2-phenylethanol) increased (Table 3). The development of VOCs during cocoa fermentation and the appropriate selection of starter culture play a crucial role especially for consumers that follow a raw-food diet. Fermented cocoa beans are a suitable food product for this new trend towards raw foods and desirable attributes should also be met after fermentation [58]. 

Despite the development of VOCs in inoculated cocoa fermentations, the volatilome profile of chocolate produced from cocoa beans originated from Brazil and Malaysia, inoculated with *S. cerevisiae* and *T. delbrueckii*, and a mixed culture of these two yeasts at the beginning of the fermentation has been assessed (Table 1) [31,34,38]. Interesting observations can be made regarding the most frequently identified VOCs in the fermented cocoa beans inoculated with yeasts and the chocolate produced also from inoculated cocoa beans with yeasts, which support the idea that some VOCs produced during fermentation can remain after processing (Figure 2). Remarkably, acetic acid was the most abundant VOC during fermentation and remained the most abundant VOC in chocolate followed by acetoin and 2-phenylethanol (Table 3) [31,34,37]. 

Several limitations were noted during the collection of the reported VOCs in both the inoculated fermented cocoa beans and the chocolate produced from different inoculated cocoa beans. The first limitation was related to the incongruency of the total number of VOCs and terpenoids reported, whereas some studies have not reported any terpenoids and the total number of VOCs identified vary from 34 to 72 compounds [6,25,31,33,34,37]. Second, studies that identified VOCs in inoculated fermented cocoa beans and chocolate are limited. Although there are no studies that have been tracking the presence of VOCs over the whole chocolate process, this review provides us with an idea of which VOCs are only formed during the fermentation of cocoa beans inoculated with yeast species and could probably remain in the end product. It is worth noting that future research in the identification of VOCs may further increase our knowledge on the role of yeasts, particularly if they increase the production of esters, aldehydes, and terpenoids. This could heighten the positive impacts of yeasts during cocoa fermentation.

#### 3.1.2. Dynamics of VOCs during Roasting

Roasting of cocoa beans is used to diminish moisture and acidity by reducing concentrations of volatile acids such as acetic acid and water [49]. However, the degree of this reduction depends on the time/temperature conditions used [59]. Several chemical reactions such as Maillard and Strecker reactions play an important role during roasting to develop the characteristic aroma and flavor of chocolate [55]. These reactions reduce sugars and amino acids to produce mainly heterocyclic groups such as aldehydes and pyrazines. Indeed, roasting has been shown to be a more effective amine generator than fermentation and it has been observed that the fermentation process supplied precursors for Strecker aldehyde formation. Overall, these reactions also depend on temperature and pH, in which higher temperatures increase amine generation [49,55,59]. 

Enormous progress is currently being made in the identification of VOCs during roasting [48,59,60,61,62,63,64,65]. In detail, a total of 243 VOCs has recently been reported from three different cocoa varieties originating from ten different countries (Table 4). The most frequently identified and abundant VOCs in roasted cocoa beans are acetic acid, 3-methylbutanoic acid, benzaldehyde, and the key aromatic compounds (2-heptanol, 2-phenylethanol, phenylacetaldehyde, and 2-methylbutanal, Table 3) [59,60,61,62,63,64]. Interestingly, we observed that the key aromatic compounds (2-phenylethyl acetate, phenylacetaldehyde, and 2-heptanol), benzaldehyde, acetic acid, as well as trimethylpirazine and 3-methylbutanal, formed during inoculated fermentations, were still present after the roasting process [48,59,60,61,62,63,64,65].

In terms of the concentration changes in VOCs during roasting, it has been shown that the key odorants formed during fermentation 2-heptanol, 2-phenylethyl acetate, 2-phenylethanol, butanoic acid, and ethyl 2-methylbutanoate remained nearly constant during the roasting process, while the formation of pyrazines, a by-product of Maillard reaction, mainly occurs during roasting [48,63,64]. It should also be noted that the loss and development of limonene, ethyl acetate, benzaldehyde, and 2-methylbutanal after thermal processing remains unclear. 

## 4. Synthesis of VOCs by Fungal Communities and their Potential Health Benefits 

Research on microbial flavor generation has tremendously increased over the last two decades and special attention was given to understand the microbial processes or microbial strategies to produce flavor compounds [25,27,66,67,68,69]. Interestingly, VOCs have been traditionally used and added to food products more for pleasure and consumers’ acceptability than for nutritional reasons. However, microorganisms and their metabolites produced have been also exploited for their tremendous potential to provide health benefits in humans. In fact, it has been recently pointed out the potential health benefits contributed mainly by VOCs in plant foods [3]. 

Volatile organic compounds can be synthesized by biological process (microorganisms during fermentation), chemical reactions (synthetic and semi-synthetic) or plant extracts, depending on the type of compound that needs to be synthesized. Concerning biological processes, the microbial metabolism includes the transformation of natural precursor (sugars, organic acids, amino acids, and fatty acids) to a wide range of flavor molecules such as aliphatics, aromatics, terpenes, lactones, O-heterocycles, and S- and N-containing compounds [68]. This review focuses on the formation of six VOCs (2-phenylethanol, phenylacetaldehyde, 2-methylbutanal, benzaldehyde, limonene, and 2-phenylethyl acetate) that are formed during inoculated cocoa fermentations with yeasts, and remain present after roasting, and the potential health benefits of these VOCs. Overall, a total of 36 fungi have been described as producers of the selected VOCs, as shown in Figure 3. In detail, *S. cerevisiae, P. anomala, H. uvarum*, *H. guilliermondii*, and *Galactomyces geotrichum* have been demonstrated to produce the majority of the selected VOCs, while a high variation between species of *Candida* and *Pichia* has been reported (Figure 3). 

Fusel alcohols are generally synthetized by the yeast’s Ehrlich pathway by the conversion of reducing sugars; this pathway contains a three-enzyme cascade that converts valine, leucine, and isoleucine into their corresponding alcohols [68]. The microbial production of L-phenylalanine to 2-phenylethanol (a rose like odor) involves the transamination of the amino acid to phenylpyruvate, decarboxylation to phenylacetaldehyde, and reduction to alcohol by yeast species [27,66,70,71,72,73,74], while the synthesis of secondary alcohols, such as 2-heptanol can be obtained from 2-heptanone (Table 5) [75]. Regarding the potential health benefits, 2-phenylethanol has been demonstrated to inhibit the growth of Gram-negative bacteria and filamentous fungi [76,77].

Besides 2-phenylethanol, other VOCs such as benzaldehyde and its derivates have been used as preservatives [85,86]. However, this compound is also able to induce antitumor activity in human cells [85,86,87,88] and antioxidant activity [53,89]. Concerning the conversion of benzyl alcohol or L-phenylalanine into benzaldehyde, this conversion has been attributed not only to yeasts but also to the basidiomycetes’ activity (Table 5) [27,66,79,80,81]. In general, aldehydes can be produced by the oxidation of alcohols such as 2-methylbutanol, 3-methylbutanol, and 2-methyl-1-propanol derived from short-chain aliphatic aldehydes such as acetaldehyde, 2-methyl-1-propanal, 2-methylbutanal, and 3-methylbutanal efficiently produced by the metabolism of yeast [66,78]. 

Regarding the biosynthesis and conversion of monoterpenes, it has been associated with the basidiomycetes’ metabolism. Limonene is produced by plants as a defense for pathogens, and this transforms into other monoterpenoids such as carvone, terpineol, perillyl alcohol, limonene epoxide, and verbenone, which can be associated with the activity of several fungal species (Table 5) [66,68,84]. Interestingly, recent *in vivo* and *in vitro* studies have reported anticarcinogenic and antinociceptive activity of limonene [90,91,92,93,94,95,96], and this compound has also been used as a preservative [97].

Last but not least, the well-known ester 2-phenylethyl acetate is recognized for its antimicrobial activity [98]. In general, the biotransformation of esters includes a more complex catabolic reaction, and it comprises the esterification of amino acids or short-chain aliphatic fatty acid and terpenyl alcohol into the desired flavor ester. The transformation of 2-phenylethyl acetate is usually metabolized from amino acids, such as phenylalanine and/or phenylpyruvic acid also from yeast species (Table 5) [66,73,82]. Besides 2-phenylethyl acetate, ethyl acetate is formed from the esterification of leucine, isoleucine or valine, and a natural aliphatic alcohol has been attributed to the activity of yeast (Table 5) [7,25,27,73,82,83]. 

Overall, the potential health effect of the selected VOCs synthesized by chemical reactions or biological processes have been linked to prevent or delay diseases or the growth of undesirable microorganisms. In summary, it has been reported from *in vivo* and *in vitro* studies the anticarcinogenic and the antinociceptive activity of limonene, [90,91,92,93,94,95,96], the antitumor activity of benzaldehyde [85,86,87,88], and antioxidant activity of benzaldehyde and its derivates [99,100]. In addition, 2-phenylethanol [76,77,101], 2-phenylethyl acetate [98], limonene [97], benzaldehydes, and derivates [89,99,100,102] have been widely used as preservatives. 

While most studies have focused on describing the capacity of VOCs to prevent, slow or inhibit the growth of microorganisms, tumors, or cells to provide health benefits, recent literature has demonstrated the capacity of these compounds to stimulate communication with the limbic system of the brain via neurons through oral routes and olfactory receptors in the nose, which changed mood and emotions by creating a sedative effect for the reduction of stress and anxiety, and finally by reducing the pain perception [103]. On the other hand, dysfunction of the chemosensory activities were highly related to differences in dietary behaviors, including loss of appetite, unintended weight loss, malnutrition, and well-known psychiatric and neurological disorders [104,105,106,107,108,109]. More important is the fact that this loss has been reported to affect the general population and it remains undiagnosed in some patients [104,110]. In this regard, 2-phenylethanol has been used to counteract the olfactory dysfunction due to multiple etiologies [111,112,113,114,115,116,117]. However, the mechanism of action of the improvements of the smell progresses and the association of chemosensory function with dietary and health outcomes remains unclear. There is no doubt that individuals with this dysfunction, highly observed in neurological diseases such as Parkinson’s, are more likely to experience a hazardous event and are the major concern for public health. Considering the positive effect of the single compounds, also produced by microbial communities during cocoa fermentation, this review hypothesizes that the consumption of chocolate produced from inoculated cocoa beans with yeasts could provide a positive health effect to consumers. However, more comprehensive studies are required to confirm the potential effect of VOCs from chocolate in human health.

In terms of international legal regulations, according to the Join FAO/WHO expert committee on food additives, all the VOCs proposed in this review are categorized as flavoring agents and do not represent a safety concern since they are predictably metabolized efficiently into innocuous products and their estimated daily intake are below the threshold for daily human intake [118].

## 5. Conclusions

Microbial communities in, on, and around our food are essential for exploring the interaction between the food system and its inter-connectedness with human health. Tracking the production of functional compounds produced by microbes will serve to improve the formation of desirable compounds. Future perspectives on the selection of the best candidate starter cultures possessing genes coding for oral usage to acquire desirable compounds and the mechanism underlying flavor perception linked to nutritional or health values need to be assessed. The findings of the present review and future analyses of VOCs may help to inform researchers, policy makers, the chocolate industry, and the general public to explore yeasts as proper producers of important VOCs to improve quality and health. 

## Figures and Tables

**Figure 1 nutrients-11-00884-f001:**
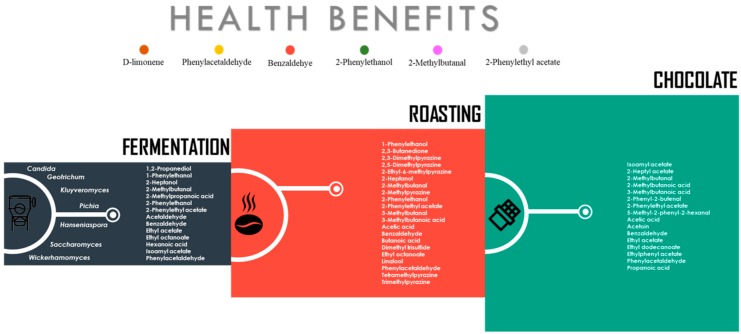
Tracking volatile compounds from chocolate.

**Figure 2 nutrients-11-00884-f002:**
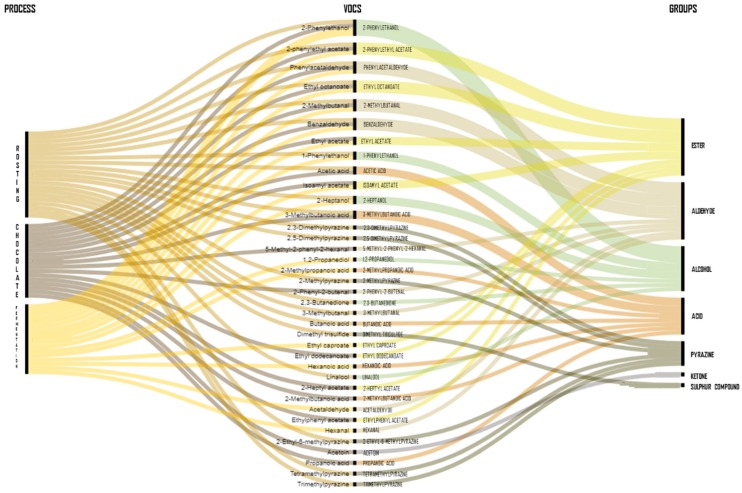
Most identified and abundant volatile compounds in fermented and roasted cocoa beans and chocolate.

**Figure 3 nutrients-11-00884-f003:**
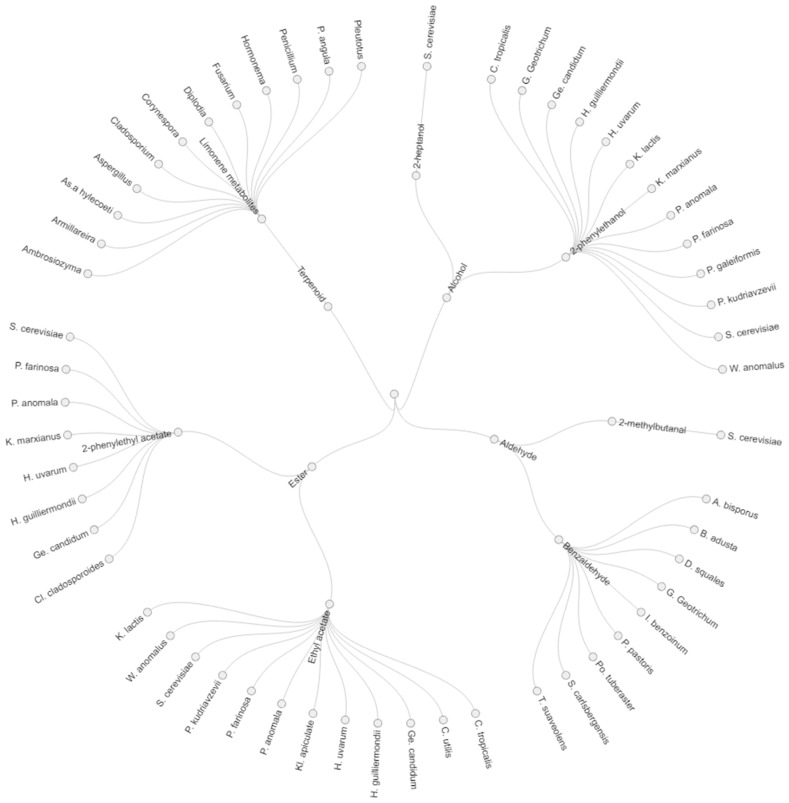
Yeast producer of selected key aromatic compounds in cocoa beans. Abbreviations: S: *Saccharomyces*, P: *Pichia*, C: *Candida*, G: *Galactomyces*, Ge: *Geotrichum*, H: *Hanseniaspora*, K: *Kluyveromyces*, W: *Wickerhamomyces,* A: *Agaricus,* B: *Bjerkandera,* D: *Dichomitus*, I: *Ischnoderma,* Po: *Polyporus,* T: *Trametes.* Kl: *Kloeckera*, Cl: *Cladosporium,* As: *Ascoide*.

**Table 1 nutrients-11-00884-t001:** Functional yeasts used as starters in cocoa fermentation.

Genera/Species	Year	Country	Type of Cocoa Bean	Type of Fermentation	Amount	VOCs		Sensorial Analysis	Reference
						F	C		
*Kluyveromyces marxianus* MMIII-41	2008	Brazil	NM	Plastic basket	45 kg	-	-	+	Leal et al., 2008 [28]
*Saccharomyces cerevisiae* UFLA CA11	2014	Brazil	PH16	Wooden box	60 kg	+	-	-	Ramos et al., 2014 [33]
*Saccharomyces cerevisiae* UFLA CA11	2014	Brazil	PS1030	Wooden box	60 kg	+	-	-	Ramos et al., 2014 [33]
*Saccharomyces cerevisiae* UFLA CA11	2014	Brazil	FA13	Wooden box	60 kg	+	-	-	Ramos et al., 2014 [33]
*Saccharomyces cerevisiae* UFLA CA11	2014	Brazil	PS1319	Wooden box	60 kg	+	-	-	Ramos et al., 2014 [33]
*Saccharomyces cerevisiae, Pichia kluyveri and Hanseniaspora uvarum*	2015	Brazil	PS1319	Wooden box	100 kg	-	-	+	Batista et al., 2015 [30]
*Candida* sp.	2015	Malaysia	NM	Basket	5 kg	-	-	-	Mahazar et al., 2015 [36]
*Saccharomyces cerevisiae* H19	2015	Malaysia	NM	Basket	50 kg	-	+	+	Meersman et al., 2016 [37]
*Saccharomyces cerevisiae* H28	2015	Malaysia	NM	Basket	50 kg	-	+	+	Mersman et al., 2016 [37]
*Saccharomyces cerevisiae* H37	2015	Malaysia	NM	Basket	50 kg	-	+	+	Mersman et al., 2016 [37]
*Saccharomyces cerevisae var. chevalieri*	2015	Indonesia	Forastero	Plastic bags	NM	-	-	-	Cempaka et al., 2014 [35]
*Saccharomyces cerevisiae*	2016	Brazil	CCN51	Wooden box	100 kg	-	+	+	Menezes et al., 2016 [34]
*Saccharomyces cerevisiae*	2016	Brazil	CEPEC2004	Wooden box	100 kg	-	+	+	Menezes et al., 2016 [34]
*Saccharomyces cerevisiae*	2016	Brazil	FA13	Wooden box	100 kg	-	+	+	Menezes et al., 2016 [34]
*Saccharomyces cerevisiae*	2016	Brazil	PS1030	Wooden box	100 kg	-	+	+	Menezes et al., 2016 [34]
*Torulaspora delbrueckii*	2017	Brazil	PS1319	Wooden box	300 kg	-	+	+	Visintin et al., 2017 [31]
*T. delbrueckii*	2017	Brazil	SJ02	Wooden box	300 kg	-	+	+	Visintin et al., 2017 [31]
*S. cerevisiae and T. delbrueckii*	2017	Brazil	PS1319	Wooden box	300 kg	-	+	+	Visintin et al., 2017 [31]
*Pichia kudriavzevii* LPB06	2017	Brazil	NM	Lab scale	400 g	+	-	-	Pereira et al., 2017 [25]
*Pichia kudriavzevii* LPB07	2017	Brazil	NM	Lab scale	400 g	+	-	-	Pereira et al., 2017 [25]
*Saccharomyces cerevisiae*	2018	Cameroon	Forastero	Wooden box	200 kg	+	-	-	Mota-Gutierrez et al., 2018 [6]
*Saccharomyces cerevisiae*	2018	Cameroon	Forastero	Heap	100 kg	+	-	-	Mota-Gutierrez et al., 2018 [6]
*Saccharomyces cerevisiae and T. delbrueckii*	2018	Cameroon	Forastero	Wooden box	200 kg	+	-	-	Mota-Gutierrez et al., 2018 [6]
*Saccharomyces cerevisiae and T. delbrueckii*	2018	Cameroon	Forastero	Heap	100 kg	+	-	-	Mota-Gutierrez et al., 2018 [6]

Abbreviations: NM, not mentioned; F, Fermented cocoa volatile compounds profile; C, Chocolate volatile compounds profile from inoculated cocoa beans; PH16 (Porto hibrido/Sao Jose da Vitoria, Brazil), PS1030 (Porto Seguro/Urucuca, Brazil), FA13 (Angola/Itahuípe Brazil), PS1319 (Bahia, Brazil), CCN51 (Ecuador), CEPEC2004 (Ilhéus/Bahia, Brazil), SJ02 (Bahia, Brazil), Witches broom- resistant varieties; +, Presence; -, Absence; VOCs, volatile organic compounds.

**Table 2 nutrients-11-00884-t002:** Nutritional composition of cocoa beans expressed as g/kg.

Source	Origin	Variety	Genetic Material	Carbohydrates				Lipids	Proteins
				Sucrose	Fructose	Glucose	Total Carbohydrates		
Afoakwa et al., 2013 [56]	Ghana	NM	Unfermented				155.00	552.00	216.00
Efraim et al., 2010 [50]	Brazil	Forastero	Unfermented					548.20	238.80
Afoakwa et al., 2013 [56]	Ghana	NM	Fermented				210.00	534.00	188.00
Efraim et al., 2010 [50]	Brazil	Forastero	Fermented					556.00	169.90
Redgwell et al., 2003 [52]	Ghana	NM	Dry cocoa beans	1.58	4.18	0.62			
Redgwell et al., 2003 [52]	Ivory Coast	NM	Dry cocoa beans	1.55	2.80	0.80			
Redgwell et al., 2003 [52]	Ecuador	NM	Dry cocoa beans	4.83	1.72	0.84			
Gu et al., 2013 [53]	Papua New Guinea	Trinitario	Roasted						458.60
Gu et al., 2013 [53]	Indonesia	Trinitario	Roasted						498.50
Gu et al., 2013 [53]	China	Trinitario	Roasted					392.40	
Gu et al., 2013 [53]	China	Trinitario	Roasted					434.40	
Redgwell et al., 2003 [52]	Ghana	NM	Roasted	1.41	0.60	0.05			
Redgwell et al., 2003 [52]	Ivory Coast	NM	Roasted	2.03	0.44	0.05			134.40
Redgwell et al., 2003 [52]	Ecuador	NM	Roasted	6.24	0.61	0.11			181.70

Abbreviations: NM, not mentioned.

**Table 3 nutrients-11-00884-t003:** Concentration ranges (µg/kg) of volatile compounds of raw, fermented, and roasted cocoa beans and chocolate.

Volatile Aroma Compounds	Raw Beans [6,33]			End of Fermentation [6,33]			Roasting [59,60,61,62,63,64]			Chocolate [31,34,37]		
Aldehydes												
2-Methylbutanal	0.70	-	1.24	0.49	-	1.46	111.00	-	4500.00	0.21	-	38.30
Acetaldehyde	0.02	-	0.85	0.00	-	0.18	285.00	-	285.00	0.60	-	41.70
Benzaldehyde	0.21	-	0.55	0.59	-	0.75	28.00	-	895.00	2.77	-	53.50
Decanal	0.03	-	0.06	0.02	-	0.04				1.00	-	1.00
Dodecanal	0.00	-	0.02	0.00	-	0.01				0.10	-	0.50
Furfural	0.00	-	0.24	0.00	-	0.25	26.00	-	87.00			
Hexanal	0.02	-	3.65	0.01	-	6.55						
Nonanal	0.14	-	0.19	0.09	-	0.14	46.00	-	46.00	0.05	-	1.52
Phenylacetaldehyde	4.06	-	6.09	3.49	-	12.37	60.00	-	5500.00	0.06	-	0.15
(E)-2-Undecenal	0.00	-	0.01	0.00	-	0.05						
2-Phenyl-2-butenal	0.00	-	0.00	0.00	-	0.05						
Alcohols												
(Z)-3-Hexen-1-ol	0.00	-	37.65	0.01	-	0.02						
1,2-Propanediol	0.00	-	0.00	0.07	-	0.35				1.10	-	1.70
1-Butanol	3.20	-	33.26	0.91	-	10.50						
1-Decanol	0.01	-	0.01	0.01	-	0.01						
1-Dodecanol	0.02	-	0.17	0.05	-	0.38						
1-Heptadecanol	0.03	-	0.10	0.06	-	0.21						
1-Heptanol	0.04	-	0.05	0.00	-	0.00				0.03	-	0.05
1-Hexanol	0.21	-	0.43	0.15	-	0.22						
1-Octanol	0.06	-	0.09	0.09	-	0.17						
1-Octen-3-ol	0.03	-	0.05	0.00	-	0.18						
1-Pentanol	0.13	-	0.83	0.07	-	0.14						
1-Phenylethanol	0.29	-	0.55	0.22	-	0.34						
1-Propanol	0.00	-	1.01	0.02	-	1.02						
2,3-Butanediol	0.00	-	9.60	0.00	-	2.07	62.00	-	356.00	35.40	-	65.35
2-Ethyl-1-hexanol	0.31	-	0.49	0.14	-	0.34				0.37	-	0.71
Furfuryl alcohol	0.00	-	0.00	0.00	-	10.71				0.49	-	0.90
2-Heptanol	0.35	-	0.54	0.00	-	8.97	32.00	-	1070.00	0.00	-	0.00
2-Hexanol	0.42	-	1.13	0.07	-	0.18						
2-Methyl-1-butanol	0.00	-	3.36	0.00	-	2.75				0.10	-	3.70
2-Methyl-1-propanol	0.00	-	0.22	0.00	-	10.33						
2-Nonanol	0.04	-	0.06	0.16	-	0.78				1.00	-	1.00
2-Pentanol	25.70	-	47.70	1.52	-	4.32				0.47	-	0.47
2-Phenylethanol	0.31	-	0.55	0.00	-	6.87	63.00	-	7500.00	3.60	-	142.00
3-Methyl-1-butanol	1.09	-	1.30	0.88	-	1.86	27.00	-	238.00	0.10	-	27.10
3-Methyl-1-pentanol	0.63	-	7.64	0.00	-	3.08						
Benzyl alcohol	0.04	-	0.05	0.03	-	0.07	104.00	-	104.00	0.20	-	0.23
Ethanol	2.17	-	3.89	1.25	-	3.81	124.00	-	124.00	4.06	-	6.71
Isobutanol	0.10	-	1.54	0.06	-	0.14						
Methanol	0.00	-	15.74	0.00	-	24.41	9068.00	-	9068.00			
(E)-3-Hexen-1-ol				0.00	-	43.75						
Acids												
2-Methylpropanoic acid	0.00	-	0.00	0.00	-	0.60	79.00	-	79.00	7.70	-	48.80
3-Methylbutanoic acid	0.05	-	0.10	3.51	-	9.20	86.00	-	9700.00	0.10	-	48.10
Acetic acid	0.68	-	1.30	4.33	-	28.40	5.60	-	330000.00	734.00	-	2555.70
Butanoic acid	0.00	-	7.36	0.00	-	13.10	21.00	-	570.00	1.30	-	2555.70
Decanoic acid	0.00	-	1.32	0.00	-	0.00						
Heptanoic acid	0.00	-	9.79	0.00	-	0.09	31.00	-	31.00			
Hexanoic acid	0.16	-	2.71	0.00	-	0.50	116.00	-	116.00	0.40	-	1.47
Nonanoic acid	0.00	-	10.28	0.00	-	0.00				0.10	-	0.10
Octanoic acid	0.03	-	0.06	0.11	-	0.27						
Ketones												
2-Heptanone	0.66	-	1.28	0.88	-	3.61	85.00	-	140.00	1.10	-	5.20
2-Pentanone	1.55	-	9.73	1.01	-	2.23						
2-Undecanone	0.04	-	0.05	0.00	-	0.03				1.00	-	1.00
Acetoin	0.38	-	0.47	1.23	-	5.98	14.00	-	1143.00	1.99	-	505.20
Acetophenone	1.17	-	3.06	0.81	-	2.31	14.00	-	225.00			
Esters												
1,2-Propanediol diacetate	6.50	-	8.11	1.21	-	2.53						
Isoamyl acetate	0.00	-	56.50	0.00	-	17.65						
2,3-Butanediol diacetate	0.15	-	0.30	0.03	-	1.20						
2-Pentanol acetate	1.42	-	2.55	1.78	-	3.93						
Diethyl malate	0.00	-	0.00	0.18	-	0.44						
Diethyl succinate	0.06	-	11.65	0.00	-	0.93						
Ethyl acetate	0.00	-	18.45	0.00	-	22.82	66.00	-	66.00	1.40	-	28.90
Ethyl benzoate	0.00	-	0.02	0.13	-	0.24	2.10	-	2.10			
Ethyl butanoate	0.26	-	3.99	0.06	-	4.18						
Ethyl caproate	0.17	-	0.22	0.43	-	0.94						
Ethyl dodecanoate	0.00	-	1.64	0.38	-	1.77	24.00	-	24.00			
Ethyl octanoate	0.00	-	0.03	0.00	-	74.29	3.30	-	143.00	0.09	-	19.10
Ethyl pyruvate	0.00	-	0.88	1.78	-	20.88						
Ethyl-o-toluate	0.00	-	0.00	0.33	-	0.62						
Furfuryl acetate	1.59	-	27.04	0.13	-	3.57						
Hexyl acetate	0.00	-	0.01	0.00	-	0.04						
Isoamyl benzoate	0.10	-	0.19	0.02	-	0.56						
Isobutyl acetate	0.14	-	1.97	0.06	-	1.98						
Methyl octanoate	0.10	-	0.15	0.00	-	0.00						
Mono-ethyl succinate	0.00	-	1.98	0.00	-	0.52						
Hexyl butanoate	0.00	-	0.05	0.00	-	0.00						
Phenyl acetate	0.00	-	0.54	0.00	-	0.14						
α-Phenylethyl acetate	0.00	-	0.45	0.17	-	0.89	34.00	-	930.00	2.60	-	37.10
Propyl acetate	0.00	-	0.09	0.00	-	1.34						
β-Phenylethyl acetate	0.03	-	0.12	0.73	-	1.69						
Terpenes												
Carveol	0.01	-	0.05	0.00	-	0.00						
(Z)-Linalool oxide pyranoid	0.10	-	0.15	0.06	-	0.17						
(Z)-Linalool oxide furanoid	0.03	-	0.10	0.00	-	0.10	21.00	-	21.00			
Nerylacetone	0.02	-	0.04	0.00	-	0.00						
Limonene	6.65	-	12.37	6.43	-	30.60						
Geraniol	0.00	-	0.31	0.00	-	0.00						
Limonene epoxide	0.29	-	0.90	0.00	-	0.02						
Sabinene	0.05	-	0.17	0.05	-	0.30						
α-Caryophyllene	0.08	-	0.09	0.07	-	0.18						
α-Citral	0.03	-	0.10	0.02	-	0.15						
α-Limonene diepoxide	0.00	-	0.02	0.00	-	0.02						
β-Caryophyllene	0.01	-	0.02	0.01	-	0.03						
β-Citronellol	0.00	-	0.00	0.00	-	0.41						
β-Myrcene	1.98	-	2.32	0.96	-	3.14	66.00	-	66.00			
(E)-β-ocimene	0.08	-	0.34	0.06	-	0.51						
Lactones												
Δ-Decalactone				0.00	-	0.20						
Other compounds												
1,1-Diethoxyethane	0.06	-	21.65	0.12	-	5.83						
o-Guaiacol	0.00	-	0.01	0.02	-	0.62	230.00	-	230.00			
Phenol	0.02	-	0.03	0.02	-	0.37	7.00	-	7.00			
*trans*-Methyl dihydrojasmonate	0.02	-	0.04	0.02	-	0.04						

Values are expressed as concentration ranges (µg/kg). Not statistical analysis was applied due to unbalanced sample size. Different color showed decrease (light blue) or increase (light green) of selected VOC concentrations.

**Table 4 nutrients-11-00884-t004:** Overview of the volatile organic compounds of roasted cocoa beans from different origins under different roasting conditions.

Source	Country	Variety	Equipment	Roasting conditions	
				Temperature (°C)	Time (min)
Bonhevi et al., 2005 [60]	Ghana, Cameroon, Ivory Coast, Brazil and Ecuador	NM	GC-MS	130	48
Ramli et al., 2006 [59]	Malaysia	NM	GC-MSD	150	30
Frauendorfer and Shieberle, 2008 [63]	Grenada	Criollo	HRGC-MS	95	14
Huang and Barringer, 2011 [61]	Ecuador	NM	SIFT-MS	150	30
Van Durme et al., 2016 [62]	Ghana and Tanzania	NM	HS-SPME-GC-MS	150	30
Magagna et al., 2018 [65]	Mexico	NM	HS-SPME-GCxGC-MS	100–130	20–40
Tan and Kerr, 2018 [64]	United States of America	Forastero	GC-MS and ANN-based-e-nose	135	0–40
Magagna et al., 2017 [48]	Ecuador and Mexico	Trinitario hybrids	GCxGC-MS, GCx2GC-MS/FID	nm	nm

Abbreviations: nm: Not mentioned, SIFT-MS: Selected ion flow tube-mass spectrometry, GC-MS: Gas chromatograph-mass spectrometer, ANN: Artificial neural network, GC-MSD: Gas chromatography-Mass selective detector, HRGC-MS: High-resolution gas chromatography-mass spectrometry, HS-SPME: Head-space solid-phase micro-extraction, FID: Flame ionization detector.

**Table 5 nutrients-11-00884-t005:** Summary table of the yeast producer of selected key aromatic compounds in cocoa beans.

Group	VOCs	Microorganism	Reference
Alcohol	2-heptanol	*Saccharomyces cerevisiae*	Cappaert and Laroche, 2004 [75]
	2-phenylethanol	*Candida tropicalis*	Koné et al., 2016 [27]
		*Galactomyces geotrichum*	Koné et al., 2016 [27]
		*Geotrichum candidum*	Janssens et al., 1992 [66]
		*Hanseniaspora guilliermondii*	Moreira et al., 2005 [73]
		*Hanseniaspora uvarum*	Moreira et al., 2005 [73]
		*Kluyveromyces lactis*	Janssens et al., 1992 [66], Fabre et al., 1997 [74]
		*Kluyveromyces marxianus*	Janssens et al., 1992 [66], Whittmann et al., 2002 [72], Etschman et al., 2005 [71], Fabre et al., 1997 [74]
		*Pichia anomala*	Janssens et al., 1992 [66]
		*Pichia farinosa*	Janssens et al., 1992 [66]
		*Pichia galeiformis*	Koné et al., 2016 [27]
		*Pichia kudriavzevii*	Koné et al., 2016 [27]
		*Saccharomyces cerevisiae*	Kim et al., 2014 [70], Koné et al., 2016 [27], Schwan and Wheals, 2004 [7], Moreira et al., 2005 [73], Fabre et al., 1997 [74]
		*Wickerhamomyces anomalus*	Koné et al., 2016 [27]
Aldehydes	2-methylbutanal	*Saccharomyces cerevisiae*	Janssens et al., 1992 [66], Larroy et al., 2002 [78]
	Benzaldehyde	*Agaricus bisporus*	Janssens et al., 1992 [66]
		*Bjerkandera adusta*	Lapadatescu et al., 1997 [79]
		*Dichomitus squales*	Lapadatescu et al., 1997 [79]
		*Galactomyces geotrichum*	Koné et al., 2016 [27]
		*Ischnoderma benzoinum*	Lapadatescu et al., 1997 [79]
		*Pichia pastoris*	Berger, 2007 [68]
		*Polyporus tuberaster*	Kawabe and Morita, 1994 [80]
		*Saccharomyces carlsbergensis*	Pal et al., 2009 [81]
	Phenylacetaldehyde	*Kluyveromyces marxianus*	Etschman et al., 2005 [71]
		*Acetobacter*	Berger, 2007 [68]
Ester	Ethyl acetate	*Candida tropicalis*	Koné et al., 2016 [27]
		*Candida utilis*	Janssens et al., 1992 [66]
		*Geotrichum candidum*	Janssens et al., 1992 [66]
		*Hanseniaspora guilliermondii*	Rojas et al., 2001 [82]
		*Hanseniaspora uvarum*	Rojas et al., 2001 [82]
		*Kloeckera apiculate*	Schwan and Wheals, 2004 [7]
		*Pichia anomala*	Janssens et al., 1992 [66], Rojas et al., 2001 [82]
		*Pichia farinosa*	Janssens et al., 1992 [66]
		*Pichia kudriavzevii*	Koné et al., 2016 [27], Pereira et al., 2017 [25]
		*Saccharomyces cerevisiae*	Janssens et al., 1992 [66], Koné et al., 2016 [27], Rojas et al., 2001 [82], Schwan and Wheals, 2004 [7]
		*Wickerhamomyces anomalus*	Koné et al., 2016 [27]
		*Kluyveromyces lactis*	Van Laere et al., 2008 [83]
	2-Phenylethyl acetate	*Cladosporium cladosporoides*	Janssens et al., 1992 [66]
		*Geotrichum candidum*	Janssens et al., 1992 [66]
		*Hanseniaspora guilliermondii*	Rojas et al., 2001 [82], Moreira et al., 2005 [73]
		*Hanseniaspora uvarum*	Rojas et al., 2001 [82]
		*Kluyveromyces marxianus*	Janssens et al., 1992 [66], Whittmann et al., 2002 [72], Etschman et al., 2005 [71]
		*Pichia anomala*	Janssens et al., 1992 [66], Rojas et al., 2001 [82]
		*Pichia farinosa*	Janssens et al., 1992 [66]
		*Saccharomyces cerevisiae*	Kone et al., 2016 [27], Rojas et al., 2001 [82]
Terpenoid	Limonene	*Ascoidea hylecoeti*	Janssens et al., 1992 [66]
	Limonene metabolites (terpineol, verbenol)	*Armillareira,* *Aspergillus* *Cladosporium*	Duetz et al., 2003 [84], Janssens et al., 1992 [66]
	Limonene metabolites (limonene-1,2-epoxide)	*Corynespora* *Diplodia*	Duetz et al., 2003 [84]
	Limonene metabolites (verbenone)	*Hormonema*	Berger, 2007 [68]
	Limonene metabolites (carvone, carveol)	*Penicillium* *Pleutotus*	Janssens et al., 1992 [66], Duetz et al., 2003 [84]
	Limonene metabolites	*Pichia angula* *Ambrosiozyma* *Fusarium*	Janssens et al., 1992 [66] Berger, 2007 [68]

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
