# Peer review of "Traceability of Functional Volatile Compounds Generated on Inoculated Cocoa Fermentation and Its Potential Health Benefits"

_nutrients, 2019, doi:10.3390/nu11040884_

Round 1

Reviewer 1 Report

"This review emphasizes the role of starter yeast in cocoa bean fermentation and the following production of specific profiles of volatile organic compounds. It concludes on the potential health benefits of those VOC remaining in the final product (chocolate). The topic is of great interest and helps to understand the large variety of flavours from cocoa originating and processed in different areas of the world.

The part on potential health benefits lacks some details such as the concentrations of VOC necessary to produce those biological benefits. This could lead to further discussion about the potential of cocoa/chocolate being beneficial to prevent some pathologies at physiological doses or the need to develop high-dose synthetic or naturally extracted isolated VOC in a pharmacological setting. While this is not the main topic of the review, the choice of the authors to extrapolate on health benefits leads to a number of questions for the biologists.

There are overall some minor spelling or grammatical mistakes that can easily be corrected. The main problem is the shift between present and past tense.
- line 47: SERVE instead of SERVED
- line 58: AIMS TO DESCRIBE
- line 59: capacity of yeast species to FORM aroma
- line 86: rare species WERE involved
- line 89: because they HAVE a positive image
- line 217: depends ON temperature and pH
- line 220: VOCs HAVE been recently reported
- line 221: cocoa varieties ORIGINATING from
- line 241: special attention WAS given
- line 244: microorganisms and THEIR metabolites have also been exploited for THEIR tremendous potential

Other comments relate to the meaning of some words where it may be unclear what message the authors want to convey
- line 71: not sure of the choice of word for CONSORTIUM...
- line 134-136: the meaning of this sentence is not clear whether the inconclusive data refer to the small number of published studies or the strength of the sensory impact
- table 2: how can the total carbohydrate be much lower than individual carbohydrates? From the Redgwell reference
- line 223: are you referring to table 4?
- line 282: are you referring to table 5?

This is not an exhaustive listing but shows the overall feeling for the manuscript. A very interesting and valuable review showing the state of the art as well as gaps in the literature that need to be addressed to make this paper even more valuable to the future reader and be a strong contribution to literature."

Author Response

Reviewer #1: General comments,

"This review emphasizes the role of starter yeast in cocoa bean fermentation and the following production of specific profiles of volatile organic compounds. It concludes on the potential health benefits of those VOC remaining in the final product (chocolate). The topic is of great interest and helps to understand the large variety of flavours from cocoa originating and processed in different areas of the world. 

The part on potential health benefits lacks some details such as the concentrations of VOC necessary to produce those biological benefits. This could lead to further discussion about the potential of cocoa/chocolate being beneficial to prevent some pathologies at physiological doses or the need to develop high-dose synthetic or naturally extracted isolated VOC in a pharmacological setting. While this is not the main topic of the review, the choice of the authors to extrapolate on health benefits leads to a number of questions for the biologists.

Thank you for appreciating our work and all the remarks done to improve our manuscript. The revised manuscript was modified according to the Reviewer’s suggestions

There are overall some minor spelling or grammatical mistakes that can easily be corrected. The main problem is the shift between present and past tense. 
- line 47: SERVE instead of SERVED
Done
- line 58: AIMS TO DESCRIBE
Done
- line 59: capacity of yeast species to FORM aroma
Done
- line 86: rare species WERE involved
Done
- line 89: because they HAVE a positive image
Done
- line 217: depends ON temperature and pH
Done
- line 220: VOCs HAVE been recently reported
Done
- line 221: cocoa varieties ORIGINATING from 
Done
- line 241: special attention WAS given
Done
- line 244: microorganisms and THEIR metabolites have also been exploited for THEIR tremendous potential
Done

Other comments relate to the meaning of some words where it may be unclear what message the authors want to convey

- line 71: not sure of the choice of word for CONSORTIUM...
Line 71 was rephrase according to reviewer’s suggestion

- line 134-136: the meaning of this sentence is not clear whether the inconclusive data refer to the small number of published studies or the strength of the sensory impact
Thank you for the remark, the authors referred to the number of the small number of published studies. This information is now available in the manuscript (L:134-135)

- table 2: how can the total carbohydrate be much lower than individual carbohydrates? From the Redgwell reference.
The authors would like to apologize because the data was shifted to the left and Total carbohydrate was not reported from Redwell et al., 2003. Changes were made according to the observation of reviewer

- line 223: are you referring to table 4?
The authors would like to clarify that this phrase refers to Table 3. This table shows the most frequently identified and abundant VOCs in roasted cocoa beans while Table 4 summarized the method used to roast cocoa beans

- line 282: are you referring to table 5?
Changes were made according to the observation of the author

This is not an exhaustive listing but shows the overall feeling for the manuscript. A very interesting and valuable review showing the state of the art as well as gaps in the literature that need to be addressed to make this paper even more valuable to the future reader and be a strong contribution to literature."

Reviewer 2 Report

Please correct some compound names.

Abstract

Phenylethyl ethanol to 1-phenylethanol or 2-phenylethanol

Phenylethylacetaldehyde to phenylacetaldehyde

Phenylethyl acetate to 1-phenylethyl acetate or 2-phenylethyl acetate

Line 170: Is phenylethyl acetate 1-phenylethyl acetate or 2-phenylethyl acetate?

Line 172: 1-butanol-3-methyl-acetate 3-hexen-1-ol: Isoamyl acetate, (Z) or (E)-3-hexen-1-ol

Table 3

Furfaral to Furfural

Trans-2-Undecenal to (E)-2-Undecenal

Alpha-Ethylidene-benzeneacetaldehyde to 2-Phenyl-2-butenal

2-Furfuryl alcohol to Furfuryl alcohol

Trans-3-Hexen-1-ol to (E)-3-Hexen-1-ol

1,2-Propanediol, diacetate to 1,2-Propanediol diacetate

1-Butanol, 3-methyl-acetate to Isoamyl acetate

2-Pentanol, acetate to 2-Pentyl acetate

Diethylsuccinate to Diethyl succinate

Ethyl butyrate to Ethyl butanoate

Ethyl-alpha-toluate to Ethyl o-toluate

n-Hexyl butanoate to Hexyl butanoate

Phenylethyl acetate to alpha-Phenylethyl acetate

Beta-phenylethyl acetate to beta-Phenylethyl acetate

Cis-Pyranoid-linalyol oxide to (Z)-Linalool oxide pyranoid

Cis-Furan linalool oxide to (Z)-Linalool oxide furanoid

Cis-Geranylacetone to Nerylacetone

Beta-trans-Ocimene to (E )-beta-ocimene

Decalactone: gamma- or delta-?

2-Butoxyethanol: Please delete.

Lines 222, 234, etc.: phenylethyl acetate: 1- or 2- ?

Line 225: 2,3,5-trimethylpyrazine to trimethylpyrazine

Line 235: ethyl-2-methylbutanoate to ethyl 2-methylbutanoate

Author Response

To the reviewer:

Thank you for your corrections, your observations helped us to improve the quality of our manuscript. Below you will find a detailed response to each of your comments. Changes in the manuscript are highlighted in the manuscript marked version

Reviewer #2: General comments,

Please correct some compound names.

Abstract

Phenylethyl ethanol to 1-phenylethanol or 2-phenylethanol Done

Phenylethylacetaldehyde to phenylacetaldehyde Done

Phenylethyl acetate to 1-phenylethyl acetate or 2-phenylethyl acetate Done

Line 170: Is phenylethyl acetate 1-phenylethyl acetate or 2-phenylethyl acetate? 2-phenylethyl acetate

Line 172: 1-butanol-3-methyl-acetate 3-hexen-1-ol: Isoamyl acetate, (Z) or (E)-3-hexen-1-ol Done

Table 3

Furfaral to Furfural Done

Trans-2-Undecenal to (E)-2-Undecenal Done

Alpha-Ethylidene-benzeneacetaldehyde to 2-Phenyl-2-butenal Done

2-Furfuryl alcohol to Furfuryl alcohol Done

Trans-3-Hexen-1-ol to (E)-3-Hexen-1-ol Done

1,2-Propanediol, diacetate to 1,2-Propanediol diacetate Done

1-Butanol, 3-methyl-acetate to Isoamyl acetate Done

2-Pentanol, acetate to 2-Pentyl acetate Done

Diethylsuccinate to Diethyl succinate Done

Ethyl butyrate to Ethyl butanoate Done

Ethyl-alpha-toluate to Ethyl o-toluate Done

n-Hexyl butanoate to Hexyl butanoate Done

Phenylethyl acetate to alpha-Phenylethyl acetate Done

Beta-phenylethyl acetate to beta-Phenylethyl acetate Done

Cis-Pyranoid-linalyol oxide to (Z)-Linalool oxide pyranoid Done

Cis-Furan linalool oxide to (Z)-Linalool oxide furanoid Done

Cis-Geranylacetone to Nerylacetone Done

Beta-trans-Ocimene to (E )-beta-ocimene Done

Decalactone: gamma- or delta-? Delta

2-Butoxyethanol: Please delete. Done

Lines 222, 234, etc.: phenylethyl acetate: 1- or 2- ? 2

Line 225: 2,3,5-trimethylpyrazine to trimethylpyrazine Done

Line 235: ethyl-2-methylbutanoate to ethyl 2-methylbutanoate Done